# Impact on Hatchability and Broiler Performance after Use of Hydrogen Peroxide Nebulization versus Formaldehyde Fumigation as Pre-Incubation Hatching Egg Disinfectants in Field Trial



Michael Pees [1,*], Gerzon Motola [2], Sarah Brüggemann-Schwarze [2], Josef Bachmeier [3], Hafez Mohamed Hafez [2] and Wiebke Tebrün [1]

1   Department of Small Mammal, Reptile and Avian Diseases, University of Veterinary Medicine Hannover, Bünteweg 9, 30559 Hanover, Germany
2   Institute of Poultry Diseases, Free University Berlin, Königsweg 63, 14163 Berlin, Germany
3   Geflügelpraxis Grüner Weg 19, 94315 Straubing, Germany
*   Correspondence: michael.pees@tiho-hannover.de

**Abstract:** Hatching egg disinfection, as part of the quality assurance system, is a standard procedure in commercial hatcheries. Formaldehyde was and is broadly used but bears high risks for the personnel. In preliminary studies, the spray application of hydrogen peroxide was successfully tested and was chosen to compare its efficacy and impact on hatchability, as well as performance during fattening, and at slaughter, to formaldehyde under field conditions. The trial was set up with hatching eggs from two breeder flocks, running parallelly in three groups ($H_2O_2$, formaldehyde and non-disinfected control) at four different flock ages (at 38, 39, 56, 57 weeks). No significant differences were noticed in the hatchery, whereas in the rearing period higher 7-day- and total mortalities occurred during trials 1 and 2 in all non-disinfected groups and one formaldehyde-treated group, making an antibiotic treatment necessary. At slaughter, the findings in all groups were comparable. Trials 3 and 4 passed without significant differences between all groups, leading to the conclusion that hatching egg disinfection lowers the risk of infection-related losses. Meanwhile, formaldehyde fumigation and the spraying of hydrogen peroxide produced similar results in all stages.

**Keywords:** broiler; hatching egg disinfection; formaldehyde; hydrogen peroxide



## 1. Introduction

Hygienic measures are taken at all stages in the broiler production chain to ensure animal health and the safety of food products. Up to 10 colony forming units (cfu) of bacteria from various genera can be isolated from the eggshell [1], however, the rearing method [2] and farm equipment [3] have an influence on the microbial load. This entails the risk of pathogens penetrating the eggshell [4,5] and infecting the embryo. A decrease in temperature after egg deposition and a humid environment are factors that facilitate bacterial penetration through the eggshell [6]. In this context, pre-incubation hatching egg disinfection is one step to control bacterial contamination and to ensure the hatching of high-quality chicks.

The most used method of hatching egg disinfectant is formaldehyde fumigation. As early as 1970, its high effectivity as pre-incubation disinfectant has been described [7]. Its effect in the hatcher was assessed [8] regarding relevant pathogens such as *Salmonella* spp. or *Pseudomonas* spp., and formaldehyde displayed its efficacy [9]. However, formaldehyde irritates the upper respiratory tract [10,11], has teratogenic and carcinogenic properties [12], and therefore it is criticized. The EU regulation 605/2014 classifies formaldehyde as carcinogenic, mutagenic, and acutely toxic. In Germany, its use is restricted to certified users under certain technical parameters only [13]; a potential market withdrawal is discussed

regularly. As substitution, various hatching egg disinfection protocols using substances such as essential oils [14], colloidal silver [15,16], electrolyzed water [17], phenols [18], UV light [19], quaternary ammonium salts [18] and ozone [20], but also different types of radiation [21–23], have been inspected for their bactericidal effectivity and their impact on production and general health parameters.

Hydrogen peroxide is one of the more extensively tested hatching egg disinfectants. Years ago, the effect of egg dipping protocols with different hydrogen peroxide concentrations were evaluated with good results. The use of 1% hydrogen peroxide via dipping led to an 85% reduction of inoculated *Salmonella* sp. in the first minute [24], with three consecutive immersions showing a 22% higher effectivity than a single immersion [25]. When using concentrations of 2% and 5% hydrogen peroxide via pressure difference dipping, no negative impact on hatching rate could be noticed [26]. Additionally, egg dipping in a 6% hydrogen peroxide solution confirmed a reduction of *Salmonella* sp. about 95% and had no adverse effects on hatchability [27]. Due to a tendency to refrain from the total wetting of the eggshell, different application methods were also tested. Compared to spraying *Salmonella* sp. inoculated eggs with 1.5% hydrogen peroxide for 10 s, an immersion in the same solution over 30 s was more effective, however the antimicrobial effect depended on the method of application [28]. When spraying the eggshell surfaces, a hydrogen peroxide concentration of 5% was necessary to reach a 5-log bacterial reduction, at the same time the hatchability increased about 2% [29]. The eggshell permeability was not measurably affected. Via the fogging of hatching eggs with a concentration of 3% hydrogen peroxide, a significant reduction of aerosol bacterial counts could be achieved, resulting in a higher loss of moisture during incubation without affecting the hatchability and subsequent broiler performance [30]. The nebulization of 6% hydrogen peroxide did not have a measurable effect on bacterial counts and lead to averaging fertility and hatching rate without a negative impact on broiler performance [31], while the use of 30% hydrogen peroxide vapor effectively decreased the microbial load by about 1 log10 cfu per egg, showing good hatchability rate, a good chick quality and good performance [31]. Melo et al. [32] confirm these results in their study using hydrogen peroxide spraying at 3%. Even though the positive aspects of hydrogen peroxide were described partly, in comparison to other disinfectants hydrogen peroxide appeared to have a low effectivity and was noticeably time dependent [33]. Different trials have examined the combination of hydrogen peroxide with other methods, e.g. UV light [34] or peracetic acid. In particular, the combination with peracetic acid has been found to be synergistic [33].

After laboratory experiments [35] with identical disinfection protocols, the aim of this study was to compare parameters concerning incubation, hatchability, health, and the performance of broiler chicks between groups of broiler-hatching eggs treated with hydrogen peroxide nebulization, formaldehyde fumigation and non-disinfected hatching eggs over four separate trials in a commercial setup. The efficacy of the disinfections procedures was verified in in vitro trials [36]. The study has been conducted as part of a German-wide project that evaluates possibilities for the reduction of multi-resistant bacteria along the poultry production chain (EsRAM, FUZ 28 177 01714).

## 2. Materials and Methods

To investigate the influence of different hatching egg disinfectants in this prospectively planned field trial, three groups were treated either with a hydrogen peroxide preparation (Wessoclean® K50 Goldline, Bio-Clean B.V., Arnhem, The Netherlands, from now on $H_2O_2$), with formaldehyde (Formaldehyd Biozid at 20%, Jäklechemie, Nuremberg, Germany, from now on FA) or without disinfectant and compared over four consecutive trials. All eggs used in this study were collected from the same two Ross 308 broiler breeder flocks (referred to as flocks A and B) at 38th, 39th, 56th and 57th week of age. Both parent flocks were managed according to the integration's guidelines, therefore receiving the same vaccination program, feed, and management. Per definition, hatching eggs that were included for this trial weighed between 48 g and 80 g. Only clean eggs were set, with a tolerance for

single dirt specks under 0.2 cm in diameter. No floor eggs or washed eggs were used. During an in-depth microbiological evaluation, a contamination between $2.26 \times 10^3$ and $1.43 \times 10^4$ cfu was detected on plate count agar, which was considered ordinary (Motola, personal communication).

Egg collection from the egg holding room on farm took place daily with proprietary trucks. The hatching eggs were not fumigated directly on farm but were stored in the hatchery at 18 °C and a maximum humidity of 70% first. As the batch of eggs for each trial covered four laying days, the first disinfection was carried out in the hatchery one to four days after lay. The disinfection, incubation and hatch always took place in the same hatchery and the disinfectants were applied before the onset of incubation as recommended by the respective companies:

Hydrogen peroxide was nebulized with specific Veugen-injectors, resulting in a particle size of 5–10 μm. The ready-to-use formula contained 0.5 mL/m$^3$ of hydrogen peroxide, 500 mL/m$^3$ of ethanol and 200 mL/m$^3$ of propan-2-ol and was applied over 1 min with 50 min exposure time afterwards. FA was used as fumigation in a concentration of 44 mL/m$^3$ applying a protocol that consisted of 15 min exposure time, 10 min neutralization time with ammonia and 300 min ventilation. The non-disinfected eggs received no mock treatment. After treatment, the eggs were held in the hatchery's egg storage room again at 18 °C and a maximum humidity of 70%. Egg age was between five and eight days. Consequently, the time elapsed between disinfection and egg setting amounted to four days. The eggs from each day of laying were distributed evenly over all groups.

For the trials 1 and 2, incubation onset was in November 2018 with an interval of one week in between the trials and 27,000 hatching eggs per group were used. The incubation of trials 3 and 4, also staggered by one week, started in March 2019 with 23,000 hatching eggs per group. After candling on day 18 of incubation and transfer to hatcher baskets, all groups received the same air sanitation treatment with Wessoclean® K50 Goldline until hatching on day 21. The incubation was realized in Single Stage Chick Master Avida setters (81,684 or 27,216 eggs) and hatchers (27,216 eggs) with the hatchery's standard incubation program. The hatched chicks from trials 1 and 2 were placed on the same farm, the chicks from trial 3 on a second farm and the chicks from trial 4 on a third farm for fattening under real-world conditions. All involved farms used the integration's commercial feed and litter and implemented site-specific management and operational processes. The licensed veterinarian and the vaccination program were the same for all farms. A partial slaughter took place at the age of 32–33 days. The final slaughter was at the age of 37–38 days according to the company's operating procedures.

For comparison between groups, parameters were evaluated at different points in the broiler production process. To evaluate the incubation period, the amounts of fertile eggs, early and late dead embryos and infertile hatching eggs detected via candling, as well as the hatching rate were recorded. During fattening, the daily and cumulative mortality rate and body mass performance were evaluated. At slaughter, the number of rejected carcasses, mean carcass weight and anomalies like runts, ascites, dermatitis, general infection, and emaciation were recorded.

Termination criteria for the study were determined beforehand. In case the daily mortality exceeding 0.2% and a necessity for antibiotic treatment, data was not statistically evaluated from this point onward. In trial 1 and 2, the termination criteria were met, which set limitations to the statistical evaluation of performance, mortality, and slaughter data. All collected data was interval-scaled and examined for normal distribution via Shapiro-Wilk-Test. Concerning candling and hatching, all four trials were evaluated together. As for fattening, trials 1 and 2 were evaluated collectively up to day 2, the trials 3 and 4 were evaluated up to slaughter. To detect differences between the three disinfection methods, the Kruskal-Wallis-Test was applied, with Mann-Whitney-U-Test as post hoc test. For the detection of differences between breeder flocks or age of the breeder flocks, the Mann-Whitney-U-Test was used directly. The evaluation was conducted with SPSS 25.0 (SPSS, IBM, Armonk, NY, USA), significance was assumed for $p \leq 0.05$.

### 3. Results

Concerning incubation parameters, the number of infertile and fertile eggs, early and late embryo mortality, and the hatching rate were almost exclusively distributed normally over all trials. However, in hatching eggs from the breeder flock B, it was noticeable that the hatching rate and number of fertile eggs was not distributed normally in the FA treated and non-disinfected groups. When comparing the different disinfectant methods, no significant differences could be observed. The mean number of fertile eggs was 92.96% in the $H_2O_2$ treated group, 92.59% in the FA treated group and 92.86% in the non-disinfected group. The mean number of early and late embryo mortality was 0.62% and 0.87% in the $H_2O_2$ treated group, 0.75% and 0.91% in the FA treated groups and 0.61% and 0.83% in the non-disinfected groups. The mean number of infertile eggs was 5.55% in the $H_2O_2$ treated group, 5.75% in the FA treated group and 5.42% in the non-disinfected group. The mean hatching rate was 89.16% for $H_2O_2$, 88.67% for FA and 88.78% for the non-disinfected eggs. For further details see Table 1.

**Table 1.** Results from the hatchery. Presentation of the number of fertile eggs (Fertile), early (Early-dead) and late dead embryos (Late-dead) and infertile eggs (Infertile) detected via candling, as well as the hatching rate (Hatch) for trials 1 to 4.

| Disinfection | Trial | Flock | Fertile [%] | Late-Dead [%] | Early-Dead [%] | Infertile [%] | Hatch [%] |
|---|---|---|---|---|---|---|---|
| $H_2O_2$ | 1 | A | 94.14 | 0.77 | 0.42 | 4.67 | 90.52 |
| $H_2O_2$ | 1 | B | 93.23 | 0.78 | 0.36 | 5.63 | 90.46 |
| Non-disinfected | 1 | A | 94.36 | 0.66 | 0.47 | 4.51 | 90.87 |
| Non-disinfected | 1 | B | 93.16 | 0.74 | 0.56 | 5.54 | 89.66 |
| FA | 1 | A | 93.70 | 0.63 | 0.66 | 5.01 | 90.11 |
| FA | 1 | B | 93.51 | 0.57 | 0.55 | 5.36 | 89.59 |
| $H_2O_2$ | 2 | A | 92.87 | 0.69 | 0.58 | 5.86 | 88.56 |
| $H_2O_2$ | 2 | B | 92.62 | 0.67 | 0.53 | 6.18 | 87.94 |
| Non-disinfected | 2 | A | 93.77 | 0.66 | 0.59 | 4.98 | 89.95 |
| Non-disinfected | 2 | B | 92.88 | 0.81 | 0.62 | 5.67 | 88.67 |
| FA | 2 | A | 92.02 | 0.92 | 0.65 | 6.41 | 88.59 |
| FA | 2 | B | 91.46 | 0.84 | 0.75 | 6.95 | 87.93 |
| $H_2O_2$ | 3 | A | 92.20 | 1.24 | 0.88 | 5.67 | 87.42 |
| $H_2O_2$ | 3 | B | 93.01 | 1.05 | 0.94 | 5.01 | 89.82 |
| Non-disinfected | 3 | A | 92.91 | 1.04 | 0.51 | 5.54 | 86.03 |
| Non-disinfected | 3 | B | 91.39 | 0.97 | 0.78 | 6.86 | 88.75 |
| FA | 3 | A | 91.83 | 1.27 | 0.96 | 5.94 | 87.26 |
| FA | 3 | B | 93.17 | 1.01 | 0.79 | 5.03 | 89.48 |
| $H_2O_2$ | 4 | A | 92.03 | 0.95 | 0.72 | 6.30 | 88.77 |
| $H_2O_2$ | 4 | B | 93.60 | 0.79 | 0.52 | 5.09 | 89.81 |
| Non-disinfected | 4 | A | 91.35 | 0.90 | 0.77 | 6.98 | 86.61 |
| Non-disinfected | 4 | B | 93.09 | 0.84 | 0.59 | 3.28 | 89.66 |
| FA | 4 | A | 91.50 | 1.05 | 0.91 | 6.54 | 86.73 |
| FA | 4 | B | 93.52 | 0.98 | 0.71 | 4.79 | 89.67 |

When comparing trials 1 and 2 with trials 3 and 4 without regard to the disinfection methods to allow for differences in age of the breeder flocks, there are significant differences concerning the number of fertile eggs ($p = 0.036$), number of early dead ($p = 0.003$) and the number of late dead ($p < 0.001$) embryos as well as the hatching rate ($p = 0.041$), as anticipated. Split up according to disinfection method, only significant differences concerning the number of late dead embryos ($p = 0.029$ in each group) remain. When comparing the breeder flocks, A and B, over all trials and disinfection methods, no significant differences were found.

In the parameters recorded during fattening and slaughter, normal distribution according to Shapiro-Wilk-Test was not confirmed for most of the parameters. In trial 1 and 2, antibiotic treatments were necessary in all non-disinfected groups and one of the FA treated groups.

Enrofloxacin (Enro-Sleecol©, Dechra, Germany) was used to treated *E. coli* infections from day 2 to day 5 of age. A statistical evaluation from day 2 onwards, including slaughter,

was refrained from as the termination criteria for the study were met. A comparison of daily mortality rates and mean body mass development on days 0 and 1 did not show significant differences between the different groups. During fattening and on slaughter, no significant differences in body weight development were seen between the groups (see Figure 1). For information, the mean cumulative mortalities on day 37 added up to 2.81% in the $H_2O_2$ treated group, 3.38% in the FA treated group and 7.09% in the non-disinfected group (see Figure 1).

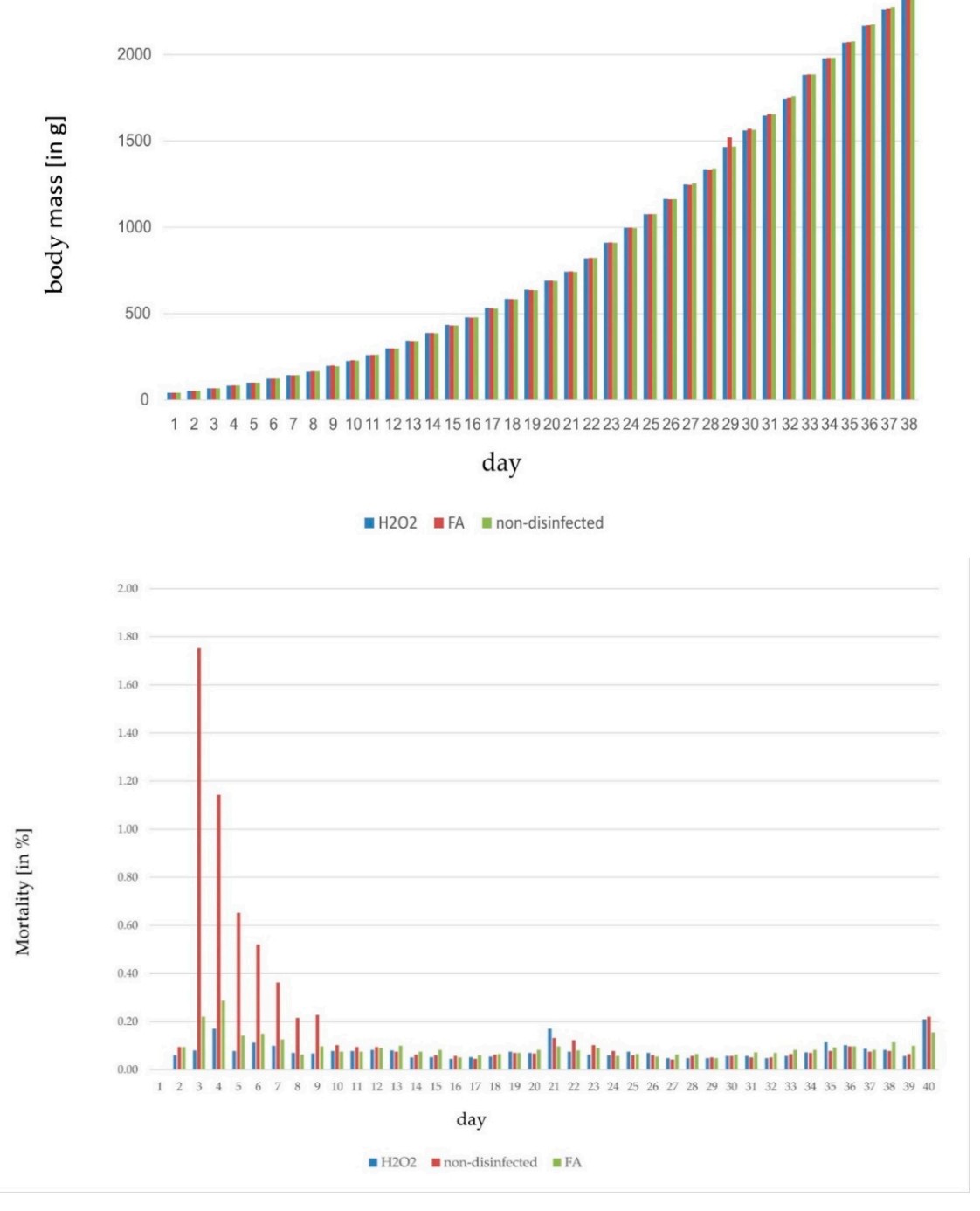

**Figure 1.** Body mass performance and daily mortality in trials 1 and 2.

Presentation of the body mass performance [g] and daily mortality [%] comparing the $H_2O_2$ treated group with the FA treated and non-disinfected group during the fattening period in trials 1 and 2.

Trials 3 and 4 did not need antibiotic treatment over the entire fattening period. An augmented mortality, as in trials 1 and 2, could not be observed, so statistical evaluation was performed for all collected data. There were no significant differences concerning daily mortality and mean body mass development between all groups (see Figure 2).

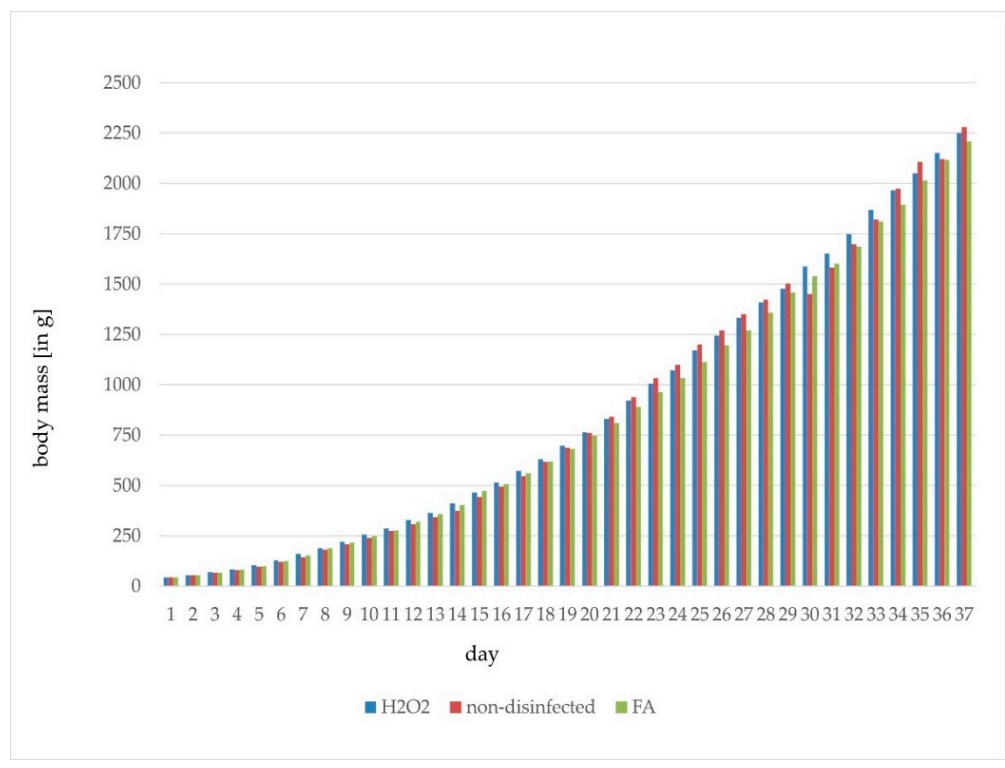

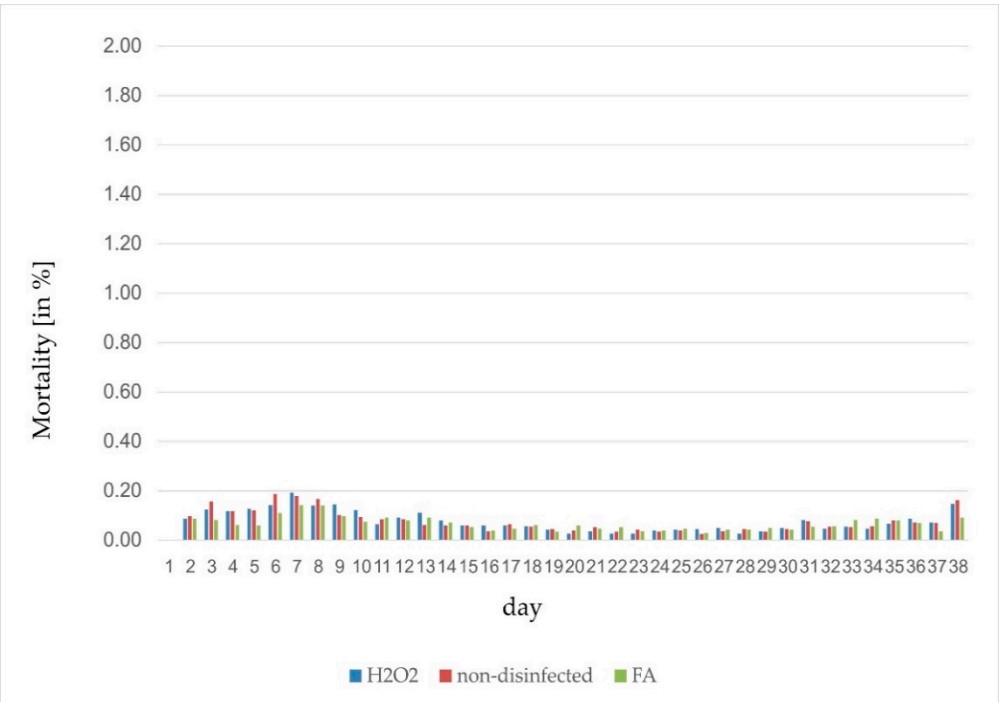

**Figure 2.** Body mass performance and mortality in trials 3 and 4.

Presentation of the body mass performance [g] and daily mortality [%] comparing the $H_2O_2$ treated group with the FA treated and non-disinfected group during the fattening period in trials 3 and 4.

In these trials, the mean cumulative mortality added up to 2.85% in the $H_2O_2$ treated group, 2.49% in the FA treated group and 2.84% in the non-disinfected group on day 37. On slaughter at 37–38 days of age, the number of registered abnormalities was not significantly different. During early slaughter on day 32–33 however, the number of animals with ascites and general infection was significantly higher in the non-disinfected group compared to the group treated with $H_2O_2$ ($p = 0.029$). For further information see Table 2.

**Table 2.** Results from the slaughterhouse. Presentation of the amount of rejected carcasses (Rej), mean body mass (BM), and the number of anomalies like runts, ascites, emaciated animals (Ema), deep dermatitis (Derm) and general infection (Gen Inf) for trials 1 to 4. The use of antibiotic treatment (AB) is shown with 1 meaning antibiotics were applied and 0 meaning no antibiotics were applied.

| Trial | Method | Flock | Age | Rej [%] | BM [g] | AB | Runts [%] | Ascites [%] | Ema [%] | Derm [%] | Gen Inf [%] |
|---|---|---|---|---|---|---|---|---|---|---|---|
| 1 | FA | A | 32 | 0.83 | 1931 | 0 | 0 | 0.18 | 0 | 0.12 | 0.42 |
| 1 | FA | B | 32 | 1.88 | 2188 | 1 | 0.39 | 0.27 | 0.09 | 0.03 | 1.04 |
| 1 | $H_2O_2$ | A | 32 | 1.41 | 1924 | 0 | 0.15 | 0.43 | 0.03 | 0.24 | 0.52 |
| 1 | $H_2O_2$ | B | 32 | 1.07 | 1863 | 0 | 0.3 | 0.27 | 0.12 | 0.09 | 0.48 |
| 1 | Non-disinfected | A | 32 | 1.39 | 1911 | 1 | 0.06 | 0.26 | 0 | 0.32 | 0.68 |
| 1 | Non-disinfected | B | 32 | 1.7 | 2219 | 1 | 0.04 | 0.63 | 0.04 | 0.26 | 0.7 |
| 1 | FA | B | 37 | 1.92 | 2418 | 1 | 0.08 | 0.47 | 0 | 0.35 | 0.55 |
| 1 | $H_2O_2$ | B | 37 | 2.15 | 2448 | 0 | 0.51 | 0.64 | 0.02 | 1.06 | 0.22 |
| 1 | Non-disinfected | B | 37 | 2.22 | 2403 | 1 | 0.25 | 0.55 | 0.01 | 0.83 | 0.41 |
| 1 | FA | A | 38 | 1.92 | 2640 | 1 | 0.07 | 0.49 | 0 | 0.99 | 0.27 |
| 1 | $H_2O_2$ | A | 38 | 2.44 | 2605 | 0 | 0.2 | 0.36 | 0 | 1.28 | 0.49 |
| 1 | Non-disinfected | A | 38 | 2 | 2583 | 1 | 0.61 | 0.82 | 0 | 0.87 | 0.2 |
| 2 | FA | A | 33 | 1.34 | 2053 | 0 | 0.05 | 0.09 | 0.09 | 0.47 | 1.14 |
| 2 | FA | B | 33 | 2.17 | 2021 | 0 | 0 | 0.22 | 0.05 | 0.16 | 1.14 |
| 2 | $H_2O_2$ | A | 33 | 2.63 | 2005 | 0 | 0.1 | 0.62 | 0.1 | 0.58 | 1.01 |
| 2 | $H_2O_2$ | B | 33 | 2.33 | 2030 | 0 | 0.06 | 0.3 | 0.03 | 0.27 | 1.2 |
| 2 | Non-disinfected | A | 33 | 1.99 | 2042 | 2 | 0.08 | 0.33 | 0.04 | 0.17 | 0.83 |
| 2 | Non-disinfected | B | 33 | 2.89 | 2072 | 2 | 0 | 0.5 | 0.04 | 0.79 | 1.21 |
| 2 | FA | A | 38 | 2.17 | 2520 | 0 | 0.09 | 0.27 | 0.06 | 0.75 | 0.75 |
| 2 | $H_2O_2$ | A | 38 | 1.94 | 2582 | 0 | 0.05 | 0.42 | 0 | 0.72 | 0.69 |
| 2 | $H_2O_2$ | B | 38 | 2.06 | 2438 | 0 | 0.01 | 0.26 | 0.02 | 0.5 | 0.99 |
| 2 | Non-disinfected | A | 38 | 2.17 | 2569 | 2 | 0.04 | 0.13 | 0.02 | 0.48 | 1.26 |
| 2 | Non-disinfected | B | 38 | 2.63 | 2532 | 2 | 0.02 | 0.38 | 0.01 | 0.53 | 1.5 |
| 2 | FA | B | 39 | 2.41 | 2639 | 0 | 0.01 | 0.63 | 0.01 | 0.62 | 0.59 |
| 3 | FA | A | 33 | 0.01 | 2031 | 0 | 0 | 0.33 | 0 | 0.87 | 0.33 |
| 3 | FA | B | 33 | 0.08 | 2007 | 0 | 6.47 | 0.43 | 0 | 1.83 | 0.22 |
| 3 | $H_2O_2$ | A | 33 | 0 | 2049 | 0 | 0.06 | 0.26 | 0 | 0.19 | 0.19 |
| 3 | $H_2O_2$ | B | 33 | 0.01 | 2042 | 0 | 0 | 0.26 | 0 | 1.03 | 0.13 |
| 3 | Non-disinfected | A | 33 | 0.02 | 2081 | 0 | 0.11 | 1.12 | 0 | 0.56 | 0.56 |
| 3 | Non-disinfected | B | 33 | 0.02 | 2070 | 0 | 0.08 | 1.17 | 0 | 0.5 | 0.5 |
| 3 | FA | B | 36 | 0.03 | 2285 | 0 | 0.04 | 0.33 | 0.02 | 2.01 | 0.58 |
| 3 | Non-disinfected | A | 36 | 0.01 | 2320 | 0 | 0.09 | 0.13 | 0.05 | 1.04 | 0.53 |
| 3 | FA | A | 37 | 0.02 | 2397 | 0 | 0.55 | 0.37 | 0.02 | 0.92 | 0.66 |
| 3 | $H_2O_2$ | A | 37 | 0.01 | 2457 | 0 | 0.11 | 0.29 | 0.01 | 0.64 | 0.42 |
| 3 | $H_2O_2$ | B | 37 | 0.03 | 2429 | 0 | 0.06 | 0.53 | 0.05 | 1.67 | 0.86 |
| 3 | Non-disinfected | B | 37 | 0.02 | 2455 | 0 | 0.25 | 0.8 | 0.01 | 0.92 | 0.28 |
| 4 | FA | A | 33 | 2.77 | 2165 | 0 | 0.28 | 2.71 | 0 | 0.06 | 0.17 |
| 4 | FA | B | 33 | 0.27 | 2117 | 0 | 0.22 | 0.32 | 0 | 0.11 | 0.11 |
| 4 | $H_2O_2$ | A | 33 | 0.28 | 2162 | 0 | 0.22 | 0.11 | 0 | 0.28 | 0.22 |

**Table 2.** *Cont.*

| Trial | Method | Flock | Age | Rej [%] | BM [g] | AB | Runts [%] | Ascites [%] | Ema [%] | Derm [%] | Gen Inf [%] |
|---|---|---|---|---|---|---|---|---|---|---|---|
| 4 | H$_2$O$_2$ | B | 33 | 1.06 | 2106 | 0 | 0.42 | 0.65 | 0 | 0.37 | 0.23 |
| 4 | Non-disinfected | A | 33 | 1.8 | 2104 | 0 | 0.18 | 1.02 | 0.05 | 0.46 | 0.55 |
| 4 | Non-disinfected | B | 33 | 1.48 | 2066 | 0 | 0.32 | 0.79 | 0 | 0.8 | 0.37 |
| 4 | FA | B | 36 | 0.59 | 2382 | 0 | 0.16 | 0.43 | 0 | 0.42 | 0.11 |
| 4 | H$_2$O$_2$ | A | 36 | 1.32 | 2405 | 0 | 0.14 | 0.7 | 0.01 | 0.65 | 0.3 |
| 4 | H$_2$O$_2$ | B | 36 | 1.43 | 2406 | 0 | 0.26 | 0.87 | 0 | 0.54 | 0.22 |
| 4 | Non-disinfected | A | 36 | 1.32 | 2392 | 0 | 0.28 | 0.76 | 0.01 | 0.7 | 0.09 |
| 4 | Non-disinfected | B | 36 | 1.68 | 2371 | 0 | 0.21 | 0.89 | 0 | 0.88 | 0.18 |
| 4 | FA | A | 37 | 1.34 | 2514 | 0 | 0.24 | 0.95 | 0 | 0.44 | 0.22 |

## 4. Discussion

This study was conducted to examine the impact of two different hatching egg disinfectants—nebulized hydrogen peroxide and formaldehyde as fumigation—compared to a non-disinfected group on general performance parameters under field conditions.

An important pre-condition for this study was to achieve an as good as possible comparability of the trials and flocks allowing direct comparison of the results. Therefore, it was ensured that conformity was given for all hatchery related parameters, as the procedures remained the same over the trials, due to using the same hatchery and their standard operation procedures. Apart from differences in the disinfection protocols, influencing factors considered were differences in eggshell thickness and possible variation in the bacterial load with respect to age and source of the breeder flocks. There were differences concerning the bacterial load, but the variation occurred in an overall low range of contamination, where none of the tested batches was considered critical. The flock identity did not have any statistically measurable impact on hatchery performance parameters, even though significant differences would not have been surprising. The occurrence of differences in hatching percentages between earlier and later production stages is well known [37,38], especially due to decreasing fertility [39] and was therefore expected.

Concerning the fattening period, the comparability of trial 1 and 2 was also given due to placement on the same farm, with e.g., comparable procedures, and bacterial preloads in the houses. For the trials 3 and 4, the birds were raised on different farms and at a later point in time, and many different factors, such as climate or management procedures, would have to be considered in cases of significant differences. The antibiotic treatment in trials 1 and 2 was a termination criterion for the trial. The data is still displayed in Figure 1 and Table 2, but statistical evaluation was not conducted with any information collected after day 2. Normally, without antibiotic intervention, the mortality rate would probably have persisted on a high level. After the antibiotic treatment however, the non-disinfected groups showed comparable daily mortality and body mass development as well as comparable findings at slaughter. Only the cumulative mortality takes account of the infectious event with augmented daily mortalities from day 2 to 7. Nevertheless, the necessity of an antibiotic treatment shows that hatching egg disinfection is an important part in the hatchery hygiene and that mortality can rise exponentially in groups without hatching egg disinfection. In in vitro tests, the use of formaldehyde and hydrogen peroxide displayed a similar disinfection efficacy [36]. While in trial 1 an antibiotic treatment in one FA treated group was necessary, all other FA groups and groups hatched from eggs treated with H$_2$O$_2$ remained unremarkable, confirming the results of the afore-mentioned in vitro tests [36]. The mean cumulative mortality was lowest for the H$_2$O$_2$ disinfection method.

In trial 3 and 4, there were no significant differences concerning the daily mortality and body mass development. However, the higher mean cumulative mortality rates matter in a commercial setup (2.49% vs. 2.85%/2.84%). Surprisingly, the non-disinfected groups also performed well and without antibiotic treatment. In comparison between the trials 1/2 and 3/4, different influencing factors such as farm, the age of the breeder flock and the

season, were observed. The general microbial load, the exposure of the hatching eggs to pathogenic bacteria, as well as the exposure of the chicks to the bacteria on farm can have an influence on the occurrence of infections.

The hatching as well as the rearing results confirm earlier reports on the safety of hydrogen peroxide when used as egg disinfection [29–31]. As the application method in former studies was different and ranged from dipping, spraying to fogging, the safety margin of $H_2O_2$ can generally be confirmed, and the nebulization method offers easy application to large egg batches. Some studies have been conducted concerning the efficacy of the treatment, also including different application routes [24,25,29,30]. Beside studies on the detection of the bacterial contamination, the efficacy was also measured using the broiler performance data [30]. In the field trials conducted in this study, we also aimed to confirm the efficacy by comparing the results to an established nebulization method and untreated groups. Therefore, the results in daily mortality and body mass development indicate that hydrogen peroxide nebulization is also an effective hatching egg disinfectant with the potential to substitute formaldehyde.

Negative control groups have repeatedly been used in experimental setups of different dimensions. Sander and Wilson [30] used a negative control group for the assessment of microbial reduction. In three trials each with 540 eggs, the negative control groups served as comparative value for hatching rates [29]. In a commercial setting, an abstention of hatching egg disinfection on 5040 hatching eggs allowed for a distinction of bacterial loads [40]. While a distinctly higher number of Enterobacteriaceae could be found in the non-disinfected group, better results concerning hatchability were recorded.

At slaughter, no statistics were obtained for trials 1 and 2, but the broilers were unremarkable and were further processed. The antibiotic treatment after the *E. coli* infection in the group without hatching egg disinfectant was effective, as mortality reached basic levels afterwards. If lesions were present in the affected birds, they were either resolved or went unnoticed upon slaughter. In trials 3 and 4, the broilers were unremarkable as well with no significant differences between groups being present.

As a field study on a large scale, a weakness of this examination is the limited amount of information on individual results and generally less detailed information in comparison to experimental studies conducted under laboratory conditions. The advantage of work under field conditions is that we could demonstrate the effect in a large farm, under economic conditions and the results derived from typical flock sizes. The results can therefore add to the experiences already published for the efficacy and safety of hydrogen peroxide in hatching egg disinfection.

## 5. Conclusions

All in all, the disinfection of hatching eggs either with $H_2O_2$ or formaldehyde fumigation demonstrated comparable results concerning hatching rates, performance and slaughter parameters. On the other hand, chicks hatched from eggs without pre-incubation disinfection have a higher risk to be subject to infectious diseases, but are not necessarily affected as multiple factors must be considered.

**Author Contributions:** Conceptualization, S.B.-S., M.P., J.B. and H.M.H.; methodology, M.P. and J.B.; software, M.P.; validation, G.M. and S.B.-S.; formal analysis, W.T.; investigation, G.M. and W.T.; resources, M.P., J.B. and H.M.H.; data curation, M.P. and W.T.; writing—original draft preparation, W.T.; writing—review and editing, W.T.; visualization, W.T.; supervision, M.P.; project administration, M.P. and H.M.H.; funding acquisition, M.P., H.M.H. and J.B. All authors have read and agreed to the published version of the manuscript.

**Funding:** The study was funded by the German Federal Office for Agriculture and Food (EsRAM, FUZ 28177 01714).

**Institutional Review Board Statement:** Ethical review and approval were waived for this study as data were recorded during normal fattening cycles without affecting or sampling the animals for study purposes.

**Informed Consent Statement:** Not applicable.

**Data Availability Statement:** Not applicable.

**Conflicts of Interest:** The authors declare no conflict of interest.

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
