# Peer review of "Impact on Hatchability and Broiler Performance after Use of Hydrogen Peroxide Nebulization versus Formaldehyde Fumigation as Pre-Incubation Hatching Egg Disinfectants in Field Trial"

_poultry, doi:10.3390/poultry2010001_

Round 1

Reviewer 1 Report

Reviewer Comments were included in the  attached file.

Author Response

Alternative methods of disinfecting eggs to the use of paraformaldehyde are very important and desirable. However, its positive effect on maintaining the health status of eggs is fundamental. In this sense, I see a lack of some analysis and informations in the manuscript.

We agree and tried to add information whenever possible. Due to the kind of the study - field research - at some point we have only limited information esp. in comparison to our preliminary experimental studies.

  1. First, information on egg weight and shell characteristics (eg surface area and thickness)is lacking, which are important as they can influence the effects of treatments.
  2.  In addition, information on egg quality was not mentioned (nest or litter eggs? clean eggs or eggs with some degree of dirt?).
    We added the information available into the text. We do not have specific information on the surface characteristics.

    3. We also verified a lack of control over the contamination degree or sanitary status of the egg used in the experiments. In our view, the authors should have included in this study analysis of the degree of contamination of the eggshells before and after the treatments, as an experimental control.
    Yes, the contamination on the egg shell was examined in a concurrent study. As the study is still to be published, we added the information as personal communication from one of the coauthors.

  3. 4. It is very important that information about the incubation conditions be informed in the Material and Methods (What was the temperature and relative humidity used in the incubation? ? Number of eggs incubated? Type of incubator? Number of eggs transferred to the hatcher? What temperature and relative humidity used in the hatcher?)
    5. What was the time interval between laying and egg collection, between laying and experimental egg disinfection procedure and between laying or disinfection and egg incubation?
    We added the general information on the system used and the procedure that was conducted. We also added the information on the time periods between laying, disinfection and incubation process.

  4. 6. Were the eggs stored or not? If so, under what temperature and humidity conditions?
    The information on storage, time intervals and climate conditions have been added.

    7. Several bird health problems were recorded in this study, but the causes of which are now known may be in the incubation environment or post-hatch housing (for example, heat stress x ascites), problems in the quality of the bedding (dermatitis), and not having exactly relationship with egg disinfection treatments.
    8. This relationship that the authors try to show in the work needs to be reassessed. For this, it is essential that the authors inform the conditions under which the eggs were incubated and the birds were housed throughout their age. Without such information, the data presented are collected without the necessary environmental controls. This undermines any conclusions about the effectiveness of using H2O2.
    We agree that more information would be desirable and for some interpretation also necessary. As a field study, the study was conducted during normal rearing cycles under the present conditions. It was therefore our main aim to make sure that the condition were identical between the methods tested. Focus was therefore on the overall development. Unfortunately, beside the information we have now included we do not have more in-dept data on the process.

Reviewer 2 Report

The manuscript entitled Impact on hatchability and broiler performance after use of hydrogen peroxide nebulization versus formaldehyde fumigation as pre-incubation hatching egg disinfectants in field trial for Poultry is very interesting and well written. With extensive field tests, it has indicated that disinfection with H2O2 demonstrated comparable results to formaldehyde fumigation concerning incubation, performance, and slaughter parameters. I believe that the paper is worthy of publication in Poultry.

Author Response

Thank you for this feedback. Indeed although field work always means that we are limited in some ways for the scientific input we also believe that the results can contribute to the field.

Reviewer 3 Report

Very interesting paper but need some more work.

Line 18 and 96 - better put age of the birds rather than week of laying

in materials nad methods - please put some information about vaccination program in all trials.

discussion chapter is too weak. - please compare your results with other authors from references. 

Author Response

Very interesting paper but need some more work.

Thank you , according to the reviewer comments we made changes and as far as possible added the required information. 

Line 18 and 96 - better put age of the birds rather than week of laying

Has been changed.

in materials nad methods - please put some information about vaccination program in all trials.

We added the information that all flocks underwent the identical standard vaccination program.

discussion chapter is too weak. - please compare your results with other authors from references. 

We have added information to the discussion chapter and now refer to earlier studies and the overall meaning - and limitations - of the study

Round 2

Reviewer 1 Report

No Comments

Reviewer 3 Report

Accept in present form